# Usability of a novel lateral flow assay for the point-of-care detection of *Neisseria gonorrhoeae*: A qualitative time-series assessment among healthcare workers in South Africa

**Lindsey de Vos**[1,☉], **Joseph Daniels**[2,☉]*, **Avuyonke Gebengu**[1], **Laura Mazzola**[3], **Birgitta Gleeson**[3], **Jérémie Piton**[3], **Mandisa Mdingi**[1], **Ranjana Gigi**[1,4], **Cecilia Ferreyra**[3], **Jeffrey D. Klausner**[5], **Remco P. H. Peters**[1,6,7]*

1 Research Unit, Foundation for Professional Development, East London, South Africa, 2 Edson College of Nursing and Health Innovation, Arizona State University, Phoenix, Arizona, United States of America, 3 Foundation for Innovative New Diagnostics, Geneva, Switzerland, 4 Institute of Social and Preventive Medicine, University of Bern, Bern, Switzerland, 5 Keck School of Medicine, University of Southern California, Los Angeles, California, United States of America, 6 Department of Medical Microbiology, University of Pretoria, Pretoria, South Africa, 7 Division of Medical Microbiology, University of Cape Town, Cape Town, South Africa

☉ These authors contributed equally to this work.
* RemcoP@foundation.co.za (RPHP); daniels.joseph@gmail.com (JD)

## Abstract

Accurate and user-friendly rapid point-of-care diagnostic tests (POCT) are needed to optimize treatment of *Neisseria gonorrhoeae*, especially in low-resource settings where syndromic management is the standard of care for sexually transmitted infections. This study aimed to assess the acceptability and usability of a novel lateral flow assay and portable reader for the point-of-care detection of *N. gonorrhoeae* infection (NG-LFA). This mixed-methods study was conducted as part of a diagnostic performance and usability evaluation of a prototype NG-LFA for detection of *N. gonorrhoeae* in symptomatic men and women at primary healthcare facilities in the Buffalo City Metro, South Africa. The Standardized System Usability Scale (SUS) was administered, and in-depth interviews were conducted among healthcare professionals (HCPs) and fieldworkers (FWs) at pre-implementation, initial use and 3- and 6-month study implementation to assess user expectations, practical experience, and future implementation considerations for the NG-LFA. Data collection and analysis was guided by the Health Technology Adoption Framework, including new health technology attributes, learnability, satisfaction, and suitability. The framework was adapted to include perceived durability. A total of 21 HCPs and FWs were trained on the NG-LFA use. SUS scores showed good to excellent acceptability ranging from 78.8–90.6 mean scores between HCPs and FWs across study time points. All transcripts were coded using Dedoose and qualitative findings were organized by learnability, satisfaction, suitability, and durability domains. Usability themes are described for each time point. Initial insecurity dissipated and specimen processing dexterity with novel POCT technology was perfected over

**Data Availability Statement:** The dataset for the System Usability Scale (SUS) has been uploaded to the Open Science Framework data repository. The identifier for this dataset is DOI 10.17605/OSF.IO/ EFVHS. The qualitative data are not publicly available due to privacy or ethical restrictions as ensured by participant informed consent and South Africa's protection of personal information act (POPIA). The relevant contact for any data inquiries and access upon reasonable request is: Freedom Mukomana <Freedomm@foundation.co.za> Data Manager, Foundation for Professional Development

**Funding:** The performance evaluation was funded by a subaward from the Global Antimicrobial Resistance Innovation Fund (GAMRIF) via FIND (recipient CF). The Foundation for Professional Development led the field evaluation in the Buffalo City Metropolitan Health District and laboratory work was conducted by the University of Pretoria. Access to the Neisseria gonorrhoeae lateral flow assay (and reader) system and test kits were sponsored by FIND as developed and provided by DCN. FIND supervised data collection procedures and manuscript draft revisions. (Funder website: https://www.gov.uk/government/groups/the-global-amr-innovation-fund). The funders had no role in study design, data collection and analysis, decision to publish, or preparation of the manuscript.

**Competing interests:** The authors have declared that no competing interests exist.

**Abbreviations:** BCM-HD, Buffalo City Metropolitan Health District; FIND, The Foundation for Innovative New Diagnostics; FPD, Foundation for Professional Development; FWs, Fieldworkers; HCPs, Healthcare professionals; HCWs, Healthcare workers; IDIs, In-depth interviews; NG-LFA, Neisseria gonorrhoeae lateral flow assay (and reader); POCT, Point-of-Care Test; STIs, Sexually transmitted infections; SUS, System Usability Scale; WHO, World Health Organization; Xpert, GeneXpert®.

time especially amongst FWs through practical learning and easy-to-use instructions (learnability). Participants experienced both positive and negative test results, yielding perceived accuracy and minimal testing challenges overall (satisfaction). By 3- and 6-month use, both HCPs and FWs found the NG-LFA convenient to use in primary health care facilities often faced with space constraints and outlined perceived benefits for patients (suitability and durability). Findings show that the NG-LFA device is acceptable and usable even amongst paraprofessionals. High SUS scores and qualitative findings demonstrate high learnability, ease-of-use and suitability that provide valuable information for first-step scale-up requirements at primary healthcare level. Minor prototype adjustments would enhance robustness and durability aspects.

# Introduction

The global burden of sexually transmitted infections (STIs), particularly *Neisseria gonorrhea*, rests mostly in low- and middle-income countries (LMICs), including African nations, with limited to no change in incidence over time [1, 2]. For *N. gonorrhoeae* alone, there are an estimated 2 million cases per year in South Africa [3]. Left untreated, *N. gonorrhoeae* can lead to infertility and pelvic inflammatory diseases in women [4]. Rapid non-molecular, point-of-care tests (POCT) have the potential to address this disparity by optimizing STI treatment, and to reduce the burden of STIs by diagnosing asymptomatic infections [4–6]. Current laboratory diagnostics are expensive, not often available and takes a day or more for return of results, delaying patient diagnosis and treatment [2, 5, 7]. Rapid diagnostics have the potential to change current syndromic management (i,e. reduce over- and undertreatment) and facilitate same-day treatment that can reduce STI prevalence [8–13]. Currently, only rapid, high-performing non-molecular POCT are available for HIV and syphilis but not for other STIs, including *N. gonorrhoeae* [14]. Developing new POCT for *N. gonorrhoeae* requires a two-pronged approach that involves device development and usability [5, 14].

Specifically, coinciding with POCT performance evaluations, usability assessments with healthcare workers (HCWs) are essential to bringing such tests into clinical practice [8, 15]. The aims of these usability assessments are to determine if HCWs can learn how to conduct the POCT efficiently and effectively as well as evaluate their perspectives of using the POCT in their clinical setting and workflow. Appropriate training and quality monitoring models for HCWs to conduct POCT need to be developed in conjunction with device development [6, 9]. Such training models can ensure testing quality across different levels of staff from paraprofessionals to medical doctors and nurses. Further, training should be evaluated and modified as a result of pilot testing for quality monitoring and to support scale-up [8, 16, 17].

Assessing POCT usability by HCWs in regards to specimen handling, processing, and interpretation of results during test development can inform training and quality monitoring for roll-out [18, 19]. Specimen collection and preparation have been shown to present the greatest challenges in the testing process [15, 20]. Conversely, specimen processing and result interpretation have been shown to be easily completed especially if the results can be objectively displayed on the device [20–23]. During usability assessments, POCT learning demonstrates how health workers that initially describe challenges with specimen preparation (e.g. pipetting etc.) become more confident with adequate training and practice over time [15, 22, 24]. Models have demonstrated that untrained clinical staff can achieve consistent and effective device use, and clinically acceptable results, with appropriate instructional material and

training [17, 25, 26]. Such task-shifting has expanded testing for HIV and has the potential to serve as a model for STI testing, accelerating prevention efforts [27].

In response to the need for *N. gonorrhoeae* POCT devices, FIND and WHO developed a target product profile (TPP) for the further development of a *N. gonorrhoeae* POCT [14]. As a result, a Lateral Flow Assay (LFA) was developed through a collaboration between FIND and DCN Diagnostics (Carlsbad, CA, USA) [28]. To support device development and feasibility, end-user experiences were assessed among two different levels of HCWs in South Africa. Using a time-series approach of interviews and surveys, we assessed changes in acceptance and usability in this setting with implications for implementation.

## Materials and methods

### Study design and setting

This mixed-methods study was conducted to assess the usability of this novel prototype *N. gonorrhoeae* lateral flow assay (NG-LFA) and reader that was developed to support *N. gonorrhoeae* testing capacity in low-resource settings [14, 28]. This evaluation was part of a cross-sectional study examining the diagnostic performance of the device [28, 29] and was conducted between November 2021 and September 2022. Usability was assessed amongst HCWs. Participants took part in a time-series assessment to outline changes in usability and acceptability at key time points for skilled learning.

HCWs using the NG-LFA device were located at five facilities in the Buffalo City Metropolitan Health District (BCM-HD) in the Eastern Cape Province, South Africa with designated STI research infrastructure and capacity (Fig 1). Facility types included public primary healthcare facilities and community health centers that serve an estimated population of ~755,000 with a population density of 298 persons/km$^2$ (census 2011) [30].

### Description NG-LFA under evaluation

At the time of the study, the NG-LFA was in assay-locked and ready for technology transfer for manufacturing scale. The NG-LFA device consists of a single-use disposable cassette using a lateral flow assay for the detection of *N. gonorrhoeae*. The lateral flow assay uses a fluorescent europium conjugate [28, 29]. Unlike most lateral flow assays, the NG-LFA test results are not visible to the naked eye. Rather, the test requires a small portable reader for the detection of fluorescence and classification of results (positive, negative, error) with test results available within 25 minutes. As part of the comprehensive field evaluation, the NG-LFA was compared to the Xpert CT/NG Assay® (Cepheid, Sunnyvale, CA) to assess diagnostic performance using fresh male urine and female vaginal swab specimens [29]. For symptomatic patients (N = 400), the NG-LFA showed a sensitivity of 91.7% and specificity of 96.3% amongst women with vaginal discharge syndrome and 96.1% sensitivity and 97.2% specificity amongst men with male urethral discharge. In case of a discordant result between the NG-LFA and Xpert®, the specimen was re-run on a new NG-LFA cassette. In case of an NG-LFA reader error, staff were instructed to check and re-insert the cassette to ensure the correct orientation.

### Materials

Specimen collection kits and set-up for NG-LFA testing included: rayon swabs for vaginal specimen collection, urine jars and transfer pipets for male first-void urine, lysis buffer bottle and disposable tubes with dropper cap, as well as the NG-LFA device and NG-LFA reader (Figs 2 and 3). The NG-LFA reader is a compact 3D-printed prototype (7 x 9 x 6.5 cm, 90 g) and includes a small battery pack/power bank (165 g) with a grey USB cable (Fig 3). Pre-made

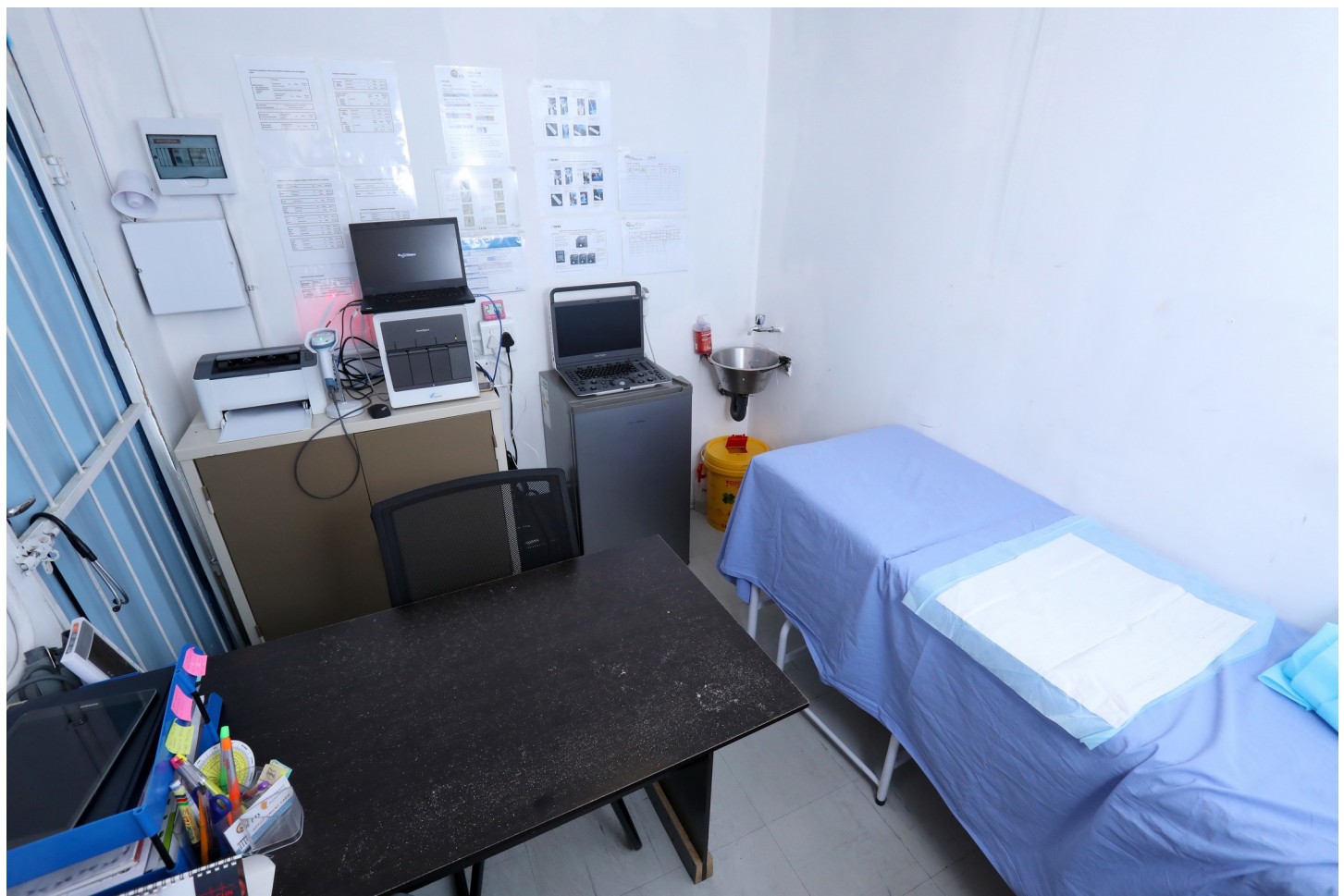

**Fig 1. STI screening infrastructure at primary healthcare.**

positive and negative control cassettes were available for quality control. The reader presents an on/off button (a green light indicates that the device is on), a Wi-Fi connectivity button (not used here), three result indicator lights (-, !, + ), and a slot for insertion of the NG-LFA cassette. Within 5 seconds of insertion, the cassette test results were indicated by a single white light on the reader, displaying whether the outcome was negative (-); positive (+); or error (!). Errors could include a mis-orientation of the cassette, device failure, or reader failure. The POCT was kept in a clear plastic container for safekeeping.

## The conceptual framework

The study was guided by the Health Technology Adoption Framework [8, 15] to assess acceptance and usability for new technology uptake in organizational settings. In this framework, acceptance and usability define feasibility. These inter-related domains consider how new devices, like NG-LFA, are perceived and believed by end-users to include trust in test results (acceptability), and then how users handle and apply the device and results (usability) [8, 15]. Taken together, these domains are further divided into six sub-domains; learnability, willingness, satisfaction, suitability, efficacy, and effectiveness [8, 19]. For this study, we adapted the framework (Fig 4) [8]. We included learnability, satisfaction, and suitability as relevant sub-

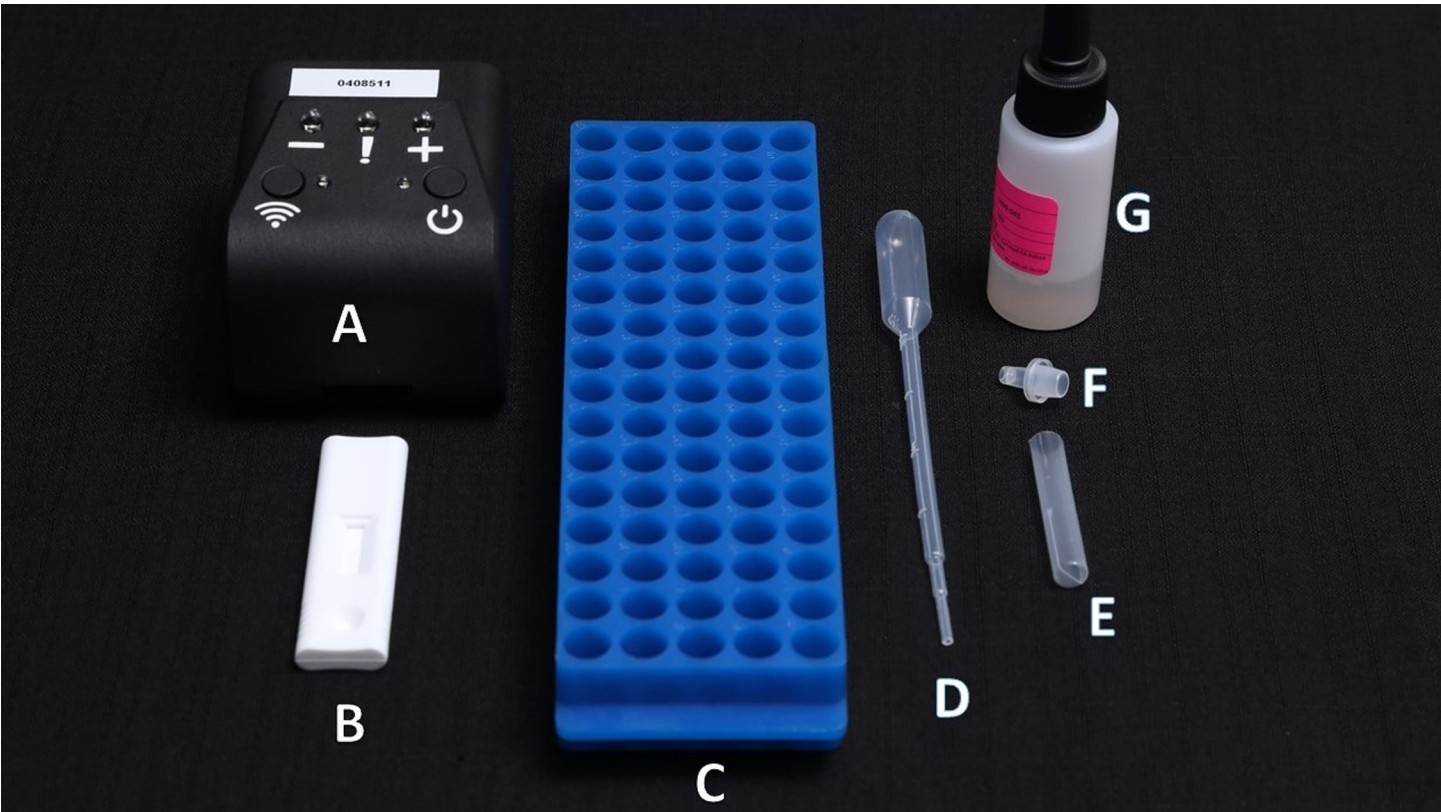

**Fig 2. FIND NG-LFA kits and testing set-up.** (A) NG-LFA Reader. (B) NG-LFA cassette. (C) Test tube rack. (D) Transfer pipette. (E) Sample tube. (F) Dropper cap. (G) Extraction buffer with medicine dropper.

domains but did not include willingness, efficacy, and effectiveness. Omitted sub-domains were not relevant to the current evaluation since we were not pilot testing for clinical protocols, and in turn, full feasibility assessment was not the study objective. However, we added durability as a domain to reflect low-resource settings where certain health system factors or challenges may impede the uptake of new technology [8], such as high patient loads, space constraints, and planned power outages (load-shedding) [31]. Further, acceptance and usability are influenced by factors relating to end-user types [8]. We, therefore, conducted a comparison between HCPs and FWs using the NG-LFA.

## Study participants

All HCW participants were either directly or indirectly involved in the implementation of the NG-LFA at any time point. These end-users included medical officers, nurses, and fieldworkers working at a primary healthcare level in Buffalo City Municipality-Health District (BCM-HD). All HCPs had 10+ years of experience in the provision of public healthcare services and patient management. All FWs had a high-school-level degree, and some had experience with prior health programs. Nearly all participants were also trained on other STI POCT as part of previous clinical research studies. Throughout data collection, every newly recruited and trained end-user was invited to participate in IDIs.

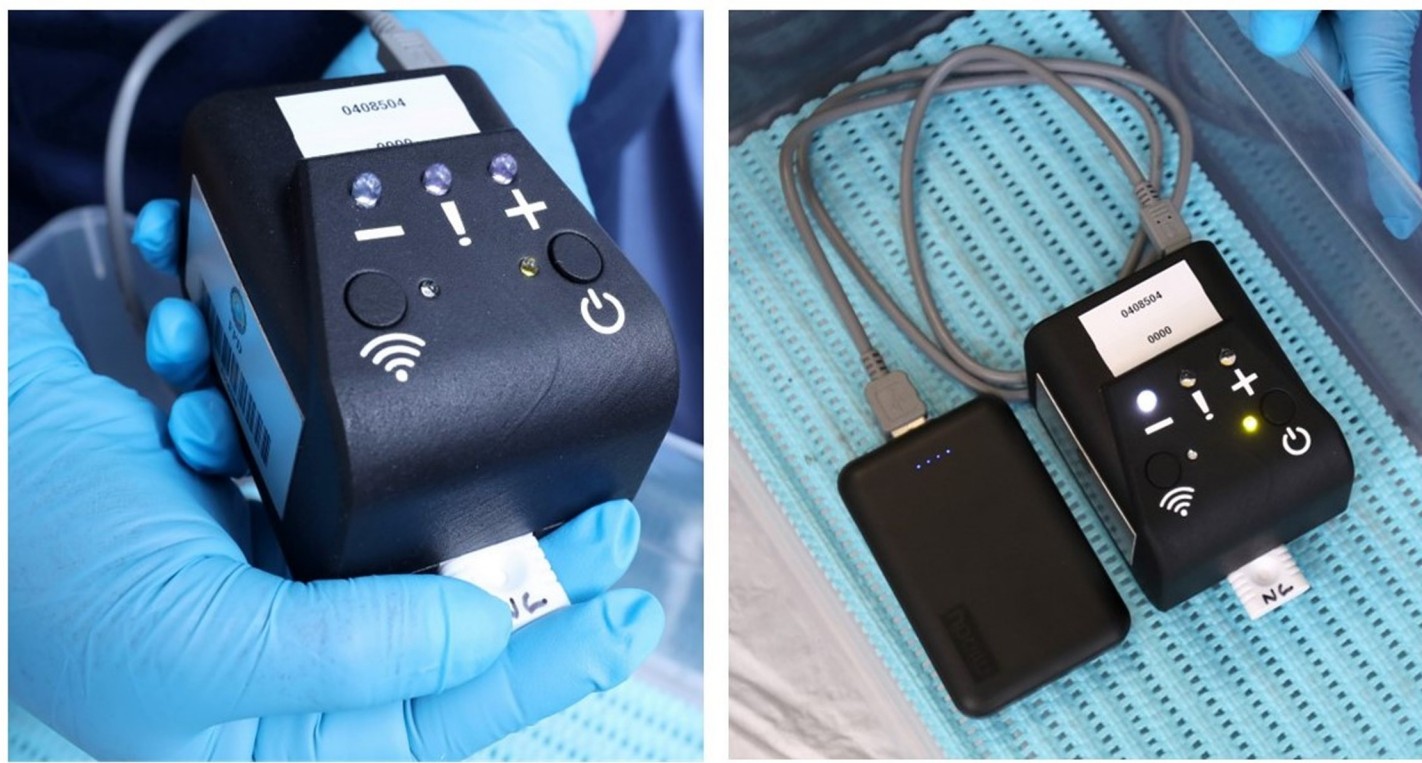

**Fig 3. NG-LFA reader and portable power bank.** Implementing staff followed each step for testing set-up through the guidance of an instruction leaflet (S1 File).

## Study procedures

Study procedures involved the following:

**Training.** All staff received a theoretical overview of the testing procedures and were given information on *N. gonorrhoeae* infection through several lectures. Staff were trained as a group on the study protocol, and the NG-LFA reader. Group training over 3 hours was conducted as a classroom set-up and was facilitated by study investigators. Staff were given

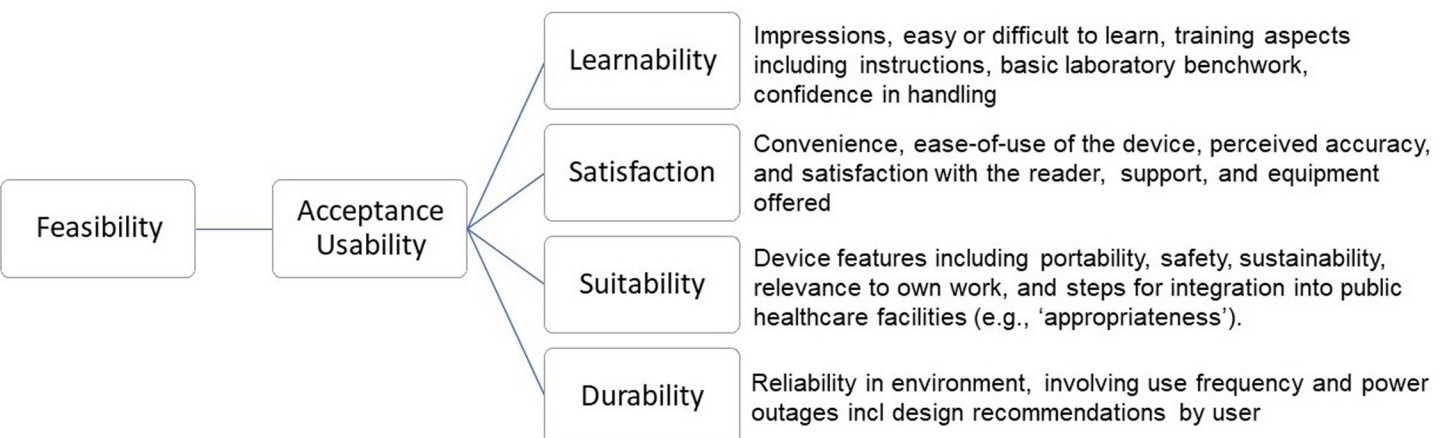

**Fig 4. Adapted health technology adoption framework for usability and acceptability sub-domains + definitions, including durability aspects (adapted from Ansbro et al., 2015).**

step-by-step instructions through quick cards (information leaflets) and supervised along the way. Staff were given all materials needed for specimen preparation and testing set-up. Staff practiced using the control cassettes, positive and negative contrived samples, sample preparation kits, and handling and interpretation of the device. Following the group training, each study site was visited by study investigators for 2–3 hours for additional training, troubleshooting, and site implementation set-up. Newly recruited staff members received a 1-day group training including a theoretical overview of the study protocol and testing procedures from senior supervisors at the office. They shadowed skilled-personnel at the facility-level for 2 days for practical NG-LFA experience. This training covered the same content as the original staff training.

**Implementation phase.** During the field evaluation, the NG-LFA testing was performed in a nurses' or fieldworkers' consulting room (either within a healthcare facility or study container, Fig 1) within a public healthcare facility. The test was made available for interested and consenting male and female patients ($\geq$ 18 years of age) presenting with male urethritis syndrome, vaginal discharge syndrome, or notified contacts of individuals with STI-associated symptoms. Participating men were requested to produce a first-void urine sample while vaginal swab specimens for NG-LFA testing were collected by nurses and then the test was either performed by the nurse or fieldworker (Fig 5). NG-LFA results were confirmed by Xpert® (Cepheid, Sunnyvale, CA). Registered nurses provided same-day test result notification, partner treatment counseling and pathogen-directed (i.e., targeted) treatment to patients that were willing to wait for the Xpert® results as the gold-standard (NG-LFA results were not used for diagnosis or treatment). If patients were not willing to wait 1.5 hours for the Xpert® results, they were given standard-of-care syndromic treatment.

## Data collection tools

**Usability surveys.** The System Usability Scale (SUS) was used to assess the usability of the NG-LFA post-training, 3- and 6-month implementation. The SUS is a validated, structured 10-item questionnaire using a 5-point Likert scale [32]. It measures perceived usability including ease of use, system satisfaction, and learnability. The SUS includes statements around technical support needs, training and perceived complexity of use [32]. Participants either agree or disagree with positively and negatively phrased statements about the device (ranging from strongly agree to strongly disagree).

**In-depth interviews.** Interviews were conducted with HCPs and FWs over four-time points and administered by a qualitative researcher (Table 1). Specific in-depth interview (IDI) protocols were developed for each study phase including specific focus areas. Questions in these time series interviews focused on end-user acceptability, experiences and usability using the Health Technology Adoption Framework [8, 15] and preferences to support potential scale-up. Each interview protocol contained a set of probes for examples and elaboration.

Table 1 shows key questions of inquiry per study phase:

## Data collection procedures

Using a purposive sampling method, participants were invited either in person, telephonically, or via email to participate in a series of surveys and interviews. Potential participants were consented using an informed consent form, before data collection. They were informed that they would be asked about their expectations, experiences, and handling of this novel NG-LFA. Study participation lasted approximately 6 months. The qualitative researcher has > 5 years of experience working on public health research studies and was trained in qualitative interviewing techniques, good clinical practice, and the study protocol. She did not have an established

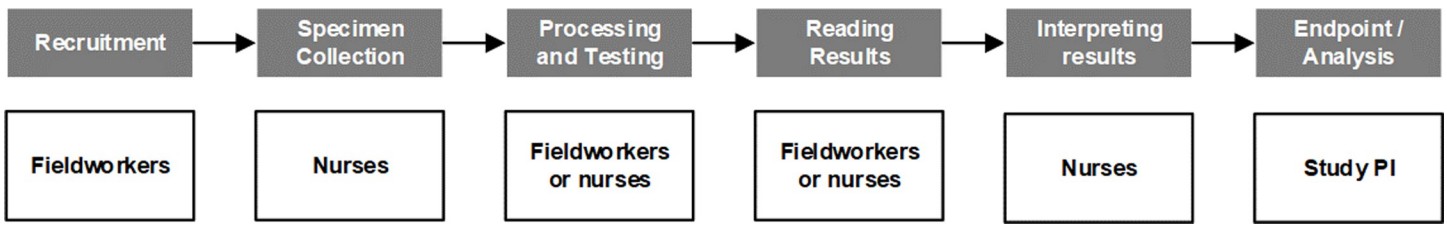

**Fig 5. NG-LFA study testing flow.**

relationship with most of the participants before the study. SUS surveys were self-administered, either on paper or through an online survey link and data was securely collected using REDCap (Research Electronic Data Capture) [33, 34]. IDIs were either conducted in person in a private location or telephonically, as scheduled with each participant. All IDIs were conducted in English and lasted approximately 40 minutes. IDIs were audio-recorded and transcribed. All transcripts were reviewed for accuracy and completion. Regular team meetings were held to discuss and refine interview guides and data collection processes.

## Data analysis

**SUS.** SUS scores were calculated and analyzed at pre-implementation, and at 3- and 6-months of implementation. Total scores were calculated using the recommended SUS scoring system [32, 35, 36]. The score contribution of each response ranged between 0–4. Five was subtracted from the sum value of the odd-numbered questions. Whereas the sum value for the even-numbered questions was subtracted from 25. These two values were added and multiplied by 2.5 to get a total score out of 100. Scores ranged between worse and best imaginable and if a SUS score is above 68, it is considered to be acceptable [35]. A mean score between 68 and 80.3 means 'good', whereas a score above 80.3 means 'excellent'. A lower score would indicate concerns with the current device. Scores were compared for HCPs and FWs at each time point.

**Qualitative.** Data were analyzed using a constant comparison approach guided by the Health Technology Adoption Framework [8, 15]. A codebook was developed by open coding a sub-set of transcripts. Codes were based on the conceptual framework. The main analytical

**Table 1. Key focus areas per study phase including relevant data collection tools.**

| Phase | Focus areas | Data collection tools |
|---|---|---|
| **1a. Training phase** | Clarity of instructions, learnability, ease of use, technical considerations | ❑ SUS |
| **1b. Study site preparation** | Expectations, proficiency, ease of use, key steps, processes for planning and implementation | ❑ IDI guide |
| **2. Initial use** | Handling, user experiences including confidence and accuracy, workflow, patient dynamics[a] including perceived trust | ❑ IDI guide |
| **3. Mid-assessment (3 months +/-)** | User experiences including functionality, suitability, patient dynamics, satisfaction, preferences | ❑ SUS and IDI guide |
| **4. Post-evaluation (6 months +/-)** | Potential implications for scale-up and sustainability including durability, changes in practice, perceived benefits | ❑ SUS and IDI guide |

[a]Patient dynamics included specimen collection and processing from male and female patients, as well as patient receptiveness to complete procedures with staff.

focus was handling, user experiences, patient-provider interactions, and clinical considerations. The codebook was applied to all transcripts by researchers JD, AG, and LDV using Dedoose [Version 9.0.17] [37]. Usability and acceptable sub-domains were compared across study phase and by user clinical role. Comparisons were drafted into a template analysis and iteratively refined through team discussions. Preliminary results were presented and discussed with the full research team. Results were organized according to the relevant attributes and themes.

## Ethics

The study was approved by the Faculty of Health Sciences Research Ethics Committee, at the University of Pretoria, South Africa (Reference No.: 510/2021). Consent was sought from all participants using an informed consent form. Recordings of interviews were voluntary. At every interview, participants received an R100 (~5.7 USD/4.7 GBP) grocery voucher for their time.

## Results

### Participants

A total of 21 HCPs and FWs (20 female and 1 male HCW) were trained on the NG-LFA and participated in the SUS post-training. The original training included 8 HCPs and 13 FWs. Several HCPs and FWs were staff integrated in antenatal care and STI screening services. Fifteen participants (n = 7 FWs, n = 8 HCPs) identified as prospective end-users participated in a pre-implementation IDI (Eligible participant flow for IDIs and SUS have been included in S2 File). As soon as staff had used the NG-LFA in practice, participants were invited to participate in an initial use IDI. At this time point, we conducted a total of 22 IDIs (n = 13 FWs, n = 9 HCPs). Fourteen participants were still involved in the study after 3 months and available for a mid-assessment IDI (n = 8 FWs, n = 6 HCPs). We interviewed a total of twelve participants after six months (n = 7 FWs, n = 6 HCPs). This resulted in a total of 63 IDIs across all time points for analysis. We calculated and compared the SUS scores for participants who evaluated the NG-LFA over 6 months.

### Usability scores

The NG-LFA device was considered acceptable at each key timepoint (Table 2). The device was deemed exceptional by HCPs at post-training and after 3-months of use (mean scores 89.9 and 90.6 respectively). Usability mean scores increased for FWs but decreased for HCPs at 6-months use (82.5 and 78.7, respectively). FWs rated the usability of the NG-LFA test as good at each time point (mean scores 79.7; 79.5; 82.5 respectively). Open comments revealed that the device was often perceived as 'straight-forward' and 'easy'. Device features, result interpretation, and step-by-step guidance defined ease-of-use. The system was further perceived as compact, portable, and yielded quick results. The overall mean SUS score for the NG-LFA device after 6-months of use was 80.6 ('excellent').

### Qualitative findings

Key qualitative findings and themes for learnability, satisfaction, suitability, and durability are presented below. Themes are described across each phase of data collection: 1) Pre-implementation: ease-of-use, dissipating health technology insecurities and benchwork training considerations; 2) Initial Use: refinement of specimen processing and perceived accuracy; 3) Mid-

**Table 2. Mean SUS scores compared across study phase and end-user category.**

| Study phase | $N^a$ | FWs SUS ($M^b$, $STD^c$) | $N^a$ | HCPs SUS ($M^b$, $STD^c$) | $N^a$ | Total SUS score ($M^b$, $STD^c$) |
|---|---|---|---|---|---|---|
| **Pre-implementation** | 13 | 79.7 (16.6) | 8 | 89.9 (7.7) | 21 | 83.5 (14.6) |
| Mid-assessment <br> • 3 months | 10 | 79.5 (12.3) | 8 | 90.6 (6.9) | 18 | 84.4 (11.4) |
| Post-evaluation <br> • 6 months | 6 | 82.5 (9.4) | 6 | 78.8 (9.8) | 12 | 80.6 (9.4) |

$^a$N = Number of respondents

$^b$M = Mean scores

$^c$STD–Standard deviation

assessment: comfort with specimen processing and design implications; and 4) Post-evaluation: information retention and perceived NG-LFA device longevity.

Illustrative quotes supporting the key findings across the time points and staff type are described in **Table 3**. We present illustrative quotes from two levels: healthcare professionals (HCPs, shaded in grey), and field workers (FWs, shaded in white).

**Pre-implementation.** *Benchwork training considerations and untested design impressions.* HCPs and FWs described the training for the NG-LFA device as helpful, with no information missing and clear, easy-to-follow steps. Although the testing process was described as such, more FWs than HCPs emphasized difficulties in using the urine transfer pipette for approximate measurements. In response to this difficulty, participants were able to practice specimen preparation several times to build confidence in specimen handling. However, both HCPS and FWs acknowledged the need for additional post-training support especially when staff are new to pipetting. Participants did not expect the light-90g weight of the NG-LFA reader prototype, which was lighter and smaller when they compared it to similar size devices that they used in the past for clinical diagnosis. Further, the size of the reader implied that it was designed to be easily portable, but some HCPs and FWs expressed concern that it would be fragile especially if used often in a setting where power outages were common. However, the supplied battery power bank and the ability to monitor battery consumption was an added benefit.

*Ease-of-use dissipates new health technology insecurities.* FWs expressed lower confidence with novel rapid tests than HCPs due to experience and the assumption that it would require more technological know-how. However, once engaged in the practical training, FWs were pleasantly surprised by the ease of use. Some HCPs also expressed anticipating advanced handling given prior experiences, but the step-by-step instructions on quick cards offered end-user guidance and enabled confidence. The learning process to use the device was described as, 'easy to read even for a layman', and 'even the field worker can do this' by FWs, and 'understand it from the word go, and 'written in simple English', by HCPs. Such that FWs were convinced the NG-LFA was not just to be administered by 'nurses'. During training, participants practiced using supplied control cassettes to assess the quality and accuracy of the device. Participants felt that the daily use of control cassettes and the availability of an 'invalid' notification on the reader also offered guidance and little room for error.

**Initial use.** *Specimen collection and processing dexterity.* Quick cards were developed that outlined steps for processing urine and vaginal swab specimens. FWs expressed initial challenges with the preparation of specimens to include getting the approximate measurements of the buffer solution using the medicine dropper, and the manual process of extracting vaginal swabs in the buffer solution. The extraction process/step of vaginal swab requires rigorous handling and squeezing of the rayon swab tip, whereby urine specimen collection and preparation for male patients was perceived as easier. After their initial testing and use of the device,

**Table 3. Themes and illustrative quotes (by end-user) organized by usability and acceptability sub-domains, interpreted across study phase.**

| Phase | Learnability | Satisfaction | Suitability | Durability |
|---|---|---|---|---|
| *Pre-Implementation* HCPs | "…I think that needs significant information [pipetting] especially if… someone has never worked with anything before… but I mean once the training is done then everyone will be able to handle it" [HCP 1] | "I did not expect it to have a plug… expected maybe, similar to HIV… I am actually impressed… because I think it gives us a preciser view… at least with the machine if there is a mistake it shows an invalid and there is a control test to check that the machine is working well…" [HCP 2] | "It's convenient, it saves space because it's not going to occupy a lot of space and it's easy to move around with as well" [HCP 3] | "I am not sure how long the paint will last, [for]… as far as weight… I love it, it is nice and light… the battery… I love that as well… it is convenient especially here in South Africa with load shedding" [HCP 2] |
| FWs | "…you get the hang of it quickly it's not complicated at all. It can look intimidating… all the things laid out in front of you… [FW 1] | "…it was more simple and easy to use. At first, I thought it would be difficult…, something only meant to be used by the research nurses…" [FW 2] | "…it's portable, and it's packed on the box, so it's easy to use it… No nothing was missing. Everything was perfectly packed on the box." [FW 3] | "I thought it was gonna be something a bit heavier than that, but I was surprised that it was light, and it was delicate… I would prefer it if there aren't many people using it…"] [FW 4] |
| *Initial use* HCPs | "I think everything was fine besides that squeezing of the swab… we are just never quite sure if we did it enough because we got told so many times that that's mainly where the mistakes happen…" [HCP 4] | "…as long as you followed the process correctly… then you get different results it's not something wrong that you did, it's the fact that confirm yourself with the ct values [on GeneXpert], "okay yeah it was a weak positive so it's likely to be a negative LFA", that helped for future tests…" [HCP 2] | "It's working very well, there is nowhere that I did encounter any problem because we explain to them [patients] why do we want urine and why do we want to take swabs you know, and then that's it, they just easily accept" [HCP 5] | "I… the design is actually nice and small, but the device seems like its fragile. Especially with the movement, up and down between the clinic and our site…" [HCP 6] |
| FWs | "it's much easier with the guys' [specimens] than with the females you have to make sure that you're thorough… trying to draw out all the specimen…" [FW 5] | "…it's a little bit confusing for me now. When the results were not the same. It happened once. Now that Dr. R explained, I get [a] little bit confident." [FW 6] | "We just show them [patients] the device and how it works and also the advantages that the turnaround time is quite quickly… and also the advantage that they get treated for the specific STI" [FW 7] | "I think it's good as it is because it is light and portable and it does not require electricity the power bank is quite strong, doing a good job…" [FW 8] |
| *3 months* HCPs | "I'm okay with the guidelines… if you're asked you can sing it off by heart, yeah the… mixing with the actual measurements…, anyone who is trained to use it can use it" [HCP 2] | "…it would be definitely [keep using] the [POCT] devices, I think they're more accurate… I think they're more helpful to provide clear guidance on what treatment to use" [HCP 7] | "For instance, if you were to move from one consulting room to another it's easy to carry and… in terms of its design um it's not so much different for instance the cassette it's very similar to what they [other participants] already orientated on…" [HCP 7] | …because the device is only for testing, just insert the cassette and take it out after reading so and it's all kept in that sealed tray you know. Yeah, the battery charge sometimes when it is low, so I don't think of any damages" [HCP 3] |
| FWs | "…it has improved a lot because at the beginning we were kind of nervous that we don't put too much cotton [from the vaginal] swab… into the specimen but now I would say I'm a bit comfortable, in fact, I am comfortable, yes" [FW 4] | "…maybe we were making mistakes [before] and all that, but um after some time using it, I think we got used to the process so there are not that many discordant [results] so I can't say there are any disadvantages I can think of" [FW 7] | "I think it's difficult when they [men] have to take swabs with the nurse otherwise taking from the urine it's very simple, they don't argue with that" [FW 6] | "…the only thing is that slot… you can see that when you put it in then it goes down, you can see that it's not going to give you the results… so then you have to take it out and put it nicely inside" [FW 3] |
| *6 months* HCPs | "You know the mixing of specimens the certain measurements that we had to mix this and that it was a lot during training… but now compared to then, ai now it's a breeze I don't want to lie, in fact, it was a breeze after the first week [HCP 2] | "I think everything there works well for me; I don't see anything that is not working well. Yes, even with specimen preparation, the things that we use to prepare the specimens and even the box, so everything is quite simple and easy to identify" [HCP 9] | "I would suggest that we try other means to test maybe we use also the urine on females if we see that this person is very dry or maybe she's got sores… so it's difficult to insert…" [HCP 9] | "…I think the size works pretty well, it is a bit light and uh almost like slippery… not in the hand, the hand is fine but on a smooth surface it can be easily thrown down…" [HCP 4] |
| FWs | "I don't even look… on the instructions on the wall… and I can even train the others since I'm working with the new nurse now, so it's much easier now than before" [FW 2] | "It's straightforward it's easy to understand, easy to read, to run, to interpret, to charge, everything about it is just a button and that's it… It felt good… it's something that is long overdue, it's needed…" [FW 1] | "It's very suitable because we put our container on a cupboard then when we have a participant, we just take our container with us, go where maybe the participant is on the youth side" [FW 6] | "I would say 5 years, the reason I'm saying that because it seems so light" [FW 2] |

some participants doubted whether their handling of the vaginal swab specimens was sufficient and questioned whether it would yield concordant results.

*Technical support and accuracy considerations.* Testing did result in a few discordant results between the NG-LFA and the Xpert device®, and for some FWs, this led to confusion. Explanations given by study investigators why the NG-LFA may yield different results, dissipated initial concerns, but this confusion rested in the limited clinical education that FWs possessed. HCPs seemed to show a deeper understanding of the reasons for discordant results (e.g., how weak positives can result in negative LFAs). For all participants, experience with sample handling procedures helped to increase perception that test results were accurate. Despite this discordancy, participants expressed the importance of testing convenience and perceived benefits for patients that the NG-LFA could yield quick results for diagnosis and treatment.

**Mid-assessment (3 months).** *Comfort with specimen processing.* The simple design and clear instructions continued to ensure end-user satisfaction. By 3 months, HCPs and FWs acknowledged increased confidence in specimen processing as compared to the start of implementation. The reader functioned as expected and staff were easily able to interpret the results using the markings on the device. Staff achieved increasing concordant results of the NG-LFA compared to Xpert, which were largely assumed by end-users to be due to enhanced specimen processing dexterity. The reader and NG results gave minimal challenges and testing experiences were similar for HCPs and FWs.

*Implications of device design.* The size and structure of the device gave participants the ability to move around and use the device between departments and shared workspaces. Furthermore, the device was easy to store, manage and maintain in a confined workspace when compared to other POCTs. Since the NG-LFA reader is a 3D printed prototype, both HCPs and FWs gave examples of minor issues experienced over the past three months with the device. These included the power on/off button falling off and a reader where it was difficult to insert the cassette. Also, some HCPs expressed that a limitation of the device is that it only tests for one STI, leaving other infections unaddressed. However, given that the feature limitations were few, and the test offered quick results and treatment guidance, staff were keen that the device could be developed to benefit patients (described in Table 3).

**Post-evaluation (6 months).** *Easy retention of information.* After 6 months of implementation, participants were able to perform the testing process without referring to the quick card instruction leaflets. FWs gave examples of being able to describe the testing process and device to new staff members and patients. New staff members were trained and shadowed skilled staff members prior to conducting testing themselves, and this adjusted model of training did not impact use. Participants also expressed confidence and trust in the test due to the concordance observed of the NG-LFA and Xpert® test results. All in all, training and practical experience was perceived as adequate for both originally trained and new HCPs and FWs. Although practical experience to refine testing dexterity and exact measurements for liquid transfer was preferred by new staff from the onset of training at the office, to compliment theoretical presentations. However, outcomes revealed that this was still adequately provided either through study-site set-up or shadowing skilled staff.

*Perceived longevity of the device.* To most participants the NG-LFA was good, but not yet excellent. The perceived longevity of the current device ranged between 9 months and 5 years due to the light weight of the prototype NG-LFA reader. Recommendations by staff for improved robustness would increase usability in high-burden settings. These included added weight to the reader and fixed features (e.g., markings, buttons and batteries). However, the compact size aided the perceived suitability to the clinical environment and movement within the room or other clinic departments. Even after 6 months of use, the prototype proved to have 'taken its' place' by the prospect of offering routine STI testing to patients. The power

bank also accommodated planned and unplanned power outages by having a long battery life as the reader required minimal electricity. However, consistent power disruptions and high patient-load required staff to be organized and charge the power bank several times a week to ensure continued clinical use.

## Discussion

We assessed if the NG-LFA test was acceptable and usable, exploring relevant Health Technology Adoption Framework attributes [8, 15] among healthcare workers involved in STI service provision. This was the first evaluation of the FIND developed NG-LFA in a LMIC setting, and a time-series, mixed methods assessment allowed us to understand end-user experience for device development and associated training needs [8, 38].

Overall usability was high with SUS score results for the NG-LFA showing comparable outcomes to other novel antigen rapid tests like for SARS-CoV-2 and optical reader prototype for any rapid POCT LFA [39–41]. By administering the SUS over several time points during the study, changes in end-user perceived skills in test performance could be assessed more accurately. Although acceptability was consistent for both groups, a slightly lower overall SUS score for HCPs during implementation was found, but we attribute this change in their shift to specimen collection and clinical management responsibilities, as well as the training model for this study.

Deploying a training model that combines classroom instruction with observed practice and supervision in the clinic for the NG-LFA proved effective with minimal refresher trainings needed. Studies show that practical training and ongoing mentoring ensure testing is performed effectively while also building sustainability of device use [8, 9, 22, 27, 42, 43]. In particular, specimen preparation is an essential step, and as in other POCT studies, we found this to be the most challenging step [20, 21]. Specifically, most participants found it easier to collect and process urine samples as compared to swabs, but over time and with support, swab adeptness improved [24]. Additionally, this training model can be complemented with resources to guide each step like printed instructions as staff move toward testing independence [15, 17, 21, 25]. Such resources reduced the number of incidents of incorrect handling and specimen transfer in our study [15, 20].

In alignment with other studies, [24, 42, 44, 45] NG-LFA device features that were highly acceptable to participants included those that supported easy interpretation of test results, quality control measures, and durability. Simple markings and lighting reduced cases of result misinterpretation (e.g. basic indicators like +/-) [24, 42, 45, 46]. An alert system and control cassettes aided quality control and test accuracy along with limited built-in software to assist easy execution of the test [24, 42]. Finally, device portability and compactness was perceived to support its durability in clinical settings with high patient volume. However, the current device tests for only one STI, gonorrhoea, and device satisfaction amongst HCWs may be higher if it was able to test for other STIs [44].

Among participants, there was a high commitment to support investment in health-improving technologies in order to improve testing and treatment outcomes in this setting [6, 9, 22, 47]. The prospect that the NG-LFA could eventually increase STI testing accessibility and provide quick results for treatment and preventative care motivated participant engagement in this study [2, 9, 15, 24–26, 47]. Further, the design of the device, particularly its size and power independence, facilitated usability and was perceived as ideal for small clinical work areas common in such settings that have a high STI burden. Such results can support antibiotic stewardship for *N. gonorrhoeae* and other STIs within health districts and

municipalities and in turn slow down the process of antimicrobial resistance development, acquisition and emergence.

### Lessons learnt

A time-series, mixed-methods approach was effective in identifying when healthcare workers become adept in the use of the NG-LFA device and testing process. This approach for assessing device usability provides information relevant to its development such as end-user expectations and training interventions needed at different timepoints to improve test performance and accuracy [8]. Also, our findings further defined durability in a low-resource setting, as a critical sub-domain for usability not previously included in the Health Technology Adoption framework [8]. Durability emphasizes the influence of organizational and environmental influences on device development such that power, storage, security and space constraints should be integrated in design and features such as the planned power outages in this setting [9, 38, 43, 45, 48].

### Implications

Our study showed that paraprofessionals can easily operate the NG-LFA and conduct effective POC STI testing as in similar studies with non-clinically trained staff [13, 25]. Therefore, task-shifting may be feasible where staff like fieldworkers can operate POC tests, which may in turn allow healthcare professionals to prioritize clinical management and treatment [6, 26]. Further, we show the effectiveness of the training model such that new staff were similarly trained by more experienced colleagues and reported comparable challenges and progress in adeptness in test performance. This suggests the adequacy of a 'train-the-trainer' approach using our training model to support feasible device scale-up [8, 17].

### Limitations

This study was implemented amongst HCWs who are clinical research staff, but they are familiar with routine practice of STI services and all had extensive nursing experience in public clinics. However, insight was gained into device suitability considerations for constrained resource settings by using the device in primary healthcare facilities. Although this study was not designed to assess full feasibility since willingness, efficacy, and effectiveness were not considered at this stage of device development; user-handling experiences and design recommendations can inform future research to identify factors influential in clinic integration like organizational readiness in using the NG-LFA device in STI care and treatment.

### Conclusion

Design of POCT for STIs must include end-user preferred features for quality control and easy readability without compromising durability that is contextually grounded. Findings show that the NG-LFA is a usable and acceptable POCT that may easily be considered for use in other LMIC's based on its characteristics. These may include provisions for commonly found setting limitations such as disrupted power supply and small workspaces. The NG-LFA features can support scale-up in clinical settings to improve access to diagnosis testing and treatment, especially in settings where significant STI burden remains unaddressed, and when misdiagnosis and antibiotic overuse can lead to antimicrobial resistance [49–51]. And, if implemented appropriately with a mentored training program, then POCT can be administered by paraprofessionals so that nurses and doctors can focus on clinical care. This may

improve time around results, appropriate use of antibiotics and reach to asymptomatic patients reducing transmission in turn.

## Supporting information

**S1 File. FIND NG-LFA testing and reader- set up instructions (QuickCards).** (PDF)

**S2 File. Study participation flow for time series assessment: SUS surveys and IDIs.** (PDF)

## Acknowledgments

DCN Diagnostics is a contract developer and manufacturer of lateral flow devices for diagnostics at the point of care. DCN performed the development and manufacturing for the prototype NG-LFA device and POC reader. This study was implemented with the permission and support of the Eastern Cape provincial and Buffalo City Metro health district departments. We would like to especially thank the Foundation for Professional Development healthcare workers for their insights and contributions towards the *Neisseria gonorrhoeae* lateral flow assay performance evaluation.

## Author Contributions

**Conceptualization:** Joseph Daniels, Cecilia Ferreyra, Remco P. H. Peters.

**Data curation:** Lindsey de Vos.

**Formal analysis:** Lindsey de Vos, Joseph Daniels, Avuyonke Gebengu.

**Funding acquisition:** Cecilia Ferreyra.

**Investigation:** Lindsey de Vos, Joseph Daniels, Avuyonke Gebengu, Remco P. H. Peters.

**Methodology:** Joseph Daniels, Remco P. H. Peters.

**Project administration:** Lindsey de Vos, Mandisa Mdingi, Ranjana Gigi.

**Resources:** Laura Mazzola, Birgitta Gleeson, Cecilia Ferreyra.

**Supervision:** Joseph Daniels, Laura Mazzola, Birgitta Gleeson, Jérémie Piton, Cecilia Ferreyra, Jeffrey D. Klausner, Remco P. H. Peters.

**Visualization:** Lindsey de Vos.

**Writing – original draft:** Lindsey de Vos, Joseph Daniels.

**Writing – review & editing:** Avuyonke Gebengu, Laura Mazzola, Birgitta Gleeson, Jérémie Piton, Mandisa Mdingi, Ranjana Gigi, Cecilia Ferreyra, Jeffrey D. Klausner, Remco P. H. Peters.

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
