## [Decision Letter · Decision Letter 0]

13 Mar 2023

PONE-D-23-01802Usability of a novel lateral flow assay for the point-of-care detection of Neisseria gonorrhoeae: a qualitative time-series assessment among healthcare workers in South AfricaPLOS ONE

Dear Dr. Peters,

Thank you for submitting your manuscript to PLOS ONE. After careful consideration, we feel that it has merit but does not fully meet PLOS ONE’s publication criteria as it currently stands. Therefore, we invite you to submit a revised version of the manuscript that addresses the points raised during the review process.

We look forward to receiving your revised manuscript.

Kind regards,

Francis Xavier Kasujja

Academic Editor

PLOS ONE

Journal Requirements:

3. We note that Figures 1, 2, 3 and S1 in your submission contain copyrighted images. All PLOS content is published under the Creative Commons Attribution License (CC BY 4.0), which means that the manuscript, images, and Supporting Information files will be freely available online, and any third party is permitted to access, download, copy, distribute, and use these materials in any way, even commercially, with proper attribution. For more information, see our copyright guidelines: http://journals.plos.org/plosone/s/licenses-and-copyright.

a. You may seek permission from the original copyright holder of Figures 1, 2, 3 and S1 to publish the content specifically under the CC BY 4.0 license. 

Reviewers' comments:

Reviewer's Responses to Questions

**Comments to the Author**

1. Is the manuscript technically sound, and do the data support the conclusions?

Reviewer #1: Yes

Reviewer #2: Yes

2. Has the statistical analysis been performed appropriately and rigorously? 

Reviewer #1: Yes

Reviewer #2: Yes

3. Have the authors made all data underlying the findings in their manuscript fully available?

Reviewer #1: Yes

Reviewer #2: Yes

4. Is the manuscript presented in an intelligible fashion and written in standard English?

Reviewer #1: Yes

Reviewer #2: Yes

5. Review Comments to the Author

Reviewer #1: This paper shows a usability test of a novel device with the issues associated with a new technology. It is interesting some of the analysis regarding the durability and the differences presented with tye gold standard, which generates some resistance of some of the Health workers involved.

Reviewer #2: I feel that that the paper is very well written and definitely the study makes a positive contribution to STI diagnosis and treatment in limited resource settings. I have a few minor comments:

1. 224- remove typo

2. 320- Please explain why there are 20 females and only 1 male. It is not also very clear how many FWs and HCPs were trained.

3. 436- What were some of the recommendations by staff to improve the device. These should be stated even at this point. I know some of these are discussed later on.

4. 464 - typo

5. Why were the FWs and HCPs trained within the same duration. The narrative clearly reveals that FWs had a harder time learning how to use the tools.

6. I feel the views of the patients would have been captured as well since this was a novel method being used in this context.

6. PLOS authors have the option to publish the peer review history of their article (what does this mean?). If published, this will include your full peer review and any attached files.

Reviewer #1: **Yes: **Andres Navarro

Reviewer #2: **Yes: **Flavia Zalwango

---

## [Author Response · Author response to Decision Letter 0]

19 Apr 2023

Thank you. Our responses to each point raised by the academic editor and reviewers has been uploaded as a Docx file under 'Response to Reviewers' 

We have also copied the responses here: 

- We appreciate the technical guidance and sharing of the manuscript style requirement guidelines. The manuscript has been amended accordingly.

- Thank you for your comment. The data underlying the SUS responses has been made available on a public repository: DOI 10.17605/OSF.IO/EFVHS. There are ethical restrictions to sharing the qualitative transcripts publicly, as per the protection of personal information (POPI) act in South Africa, and informed consent as given by participants to ensure responses are kept confidential and the identity of staff are protected. This data will be made available upon reasonable request to the corresponding author. We have included the DOI and data availability statement in the cover letter. 

3. We note that Figures 1, 2, 3 and S1 in your submission contain copyrighted images. All PLOS content is published under the Creative Commons Attribution License (CC BY 4.0), which means that the manuscript, images, and Supporting Information files will be freely available online, and any third party is permitted to access, download, copy, distribute, and use these materials in any way, even commercially, with proper attribution. For more information, see our copyright guidelines: http://journals.plos.org/plosone/s/licenses-and-copyright.

- Thank you for the information regarding copyrighted images. 

Written permission has been sought and granted by the copyright holder – DCN for S1 in our submission. The QuickCards were specifically developed for this reader prototype and NG-LFA test. A figure caption has been included in the Supplementary file. 

Figures 1, 2, and 3 are under the copyright of the Foundation for Professional Development (main affiliation of first and corresponding author), we sought and received organizational approval. 

The Content Permission forms have been uploaded as “Other” as part of this submission. 

- We have reviewed our reference list. Minor edits were made but we did not make any changes to the included reference list as per the original submission.

5. Reviewer #1: This paper shows a usability test of a novel device with the issues associated with a new technology. It is interesting some of the analysis regarding the durability and the differences presented with tye gold standard, which generates some resistance of some of the Health workers involved

- Thank you very much for your review and interest in this paper.

6. Reviewer #2: I feel that that the paper is very well written and definitely the study makes a positive contribution to STI diagnosis and treatment in limited resource settings. I have a few minor comments:

- Thank you very much for your comments

1. 224- remove typo

- We have removed the additional ‘the’ (typo) from the manuscript

Line 222

2. 320- Please explain why there are 20 females and only 1 male. It is not also very clear how many FWs and HCPs were trained.

- We have added additional information to explain the difference in female and male staff members. Previous staff offered antenatal care STI screening services at collaborating healthcare facilities

Line 319-320

The original training included 8 HCPs and 13 FWs. Line 319

3. 436- What were some of the recommendations by staff to improve the device. These should be stated even at this point. I know some of these are discussed later on.

- We have included more details regarding staff recommendations in line 437-438. These included added weight and more robust features for the reader. 

4. 464 – typo - Thank you. We have amended the sentence.

Line 465

5. Why were the FWs and HCPs trained within the same duration. The narrative clearly reveals that FWs had a harder time learning how to use the tools.

- FWs and HCPs had similar responsibilities for NG-LFA specimen preparation and testing. Both staff groups were trained in a classroom/workshop set-up.

6. I feel the views of the patients would have been captured as well since this was a novel method being used in this context.

- Thank you for this comment. This manuscript focuses on usability and learnability aspects for end-users. We are actually in the process of analyzing the provider-patient experiences as perceived by staff during implementation. For future studies we aim to assess the values and preferences for POCT on multiple levels, including assessing clinical readiness and acceptance.

---

## [Editor Report · Decision Letter 1]

22 May 2023

Usability of a novel lateral flow assay for the point-of-care detection of Neisseria gonorrhoeae: a qualitative time-series assessment among healthcare workers in South Africa

PONE-D-23-01802R1

Dear Dr. Peters,

We’re pleased to inform you that your manuscript has been judged scientifically suitable for publication and will be formally accepted for publication once it meets all outstanding technical requirements.

Kind regards,

Francis Xavier Kasujja

Academic Editor

PLOS ONE
---

## [Editor Report · Acceptance letter]

24 May 2023

PONE-D-23-01802R1 

Usability of a novel lateral flow assay for the point-of-care detection of *Neisseria gonorrhoeae:* a qualitative time-series assessment among healthcare workers in South Africa 

Dear Dr. Peters:

I'm pleased to inform you that your manuscript has been deemed suitable for publication in PLOS ONE. Congratulations! Your manuscript is now with our production department. 

Kind regards, 

on behalf of

Dr. Francis Xavier Kasujja 

Academic Editor

PLOS ONE